

# Radiation dose-response (a Bayesian model) in the radiotherapy of the localized prostatic adenocarcinoma: the reliability of PSA slope changes as a response surrogate endpoint

Reza Ali Mohammadpour[1], Jamshid Yazdani- Charati[1], SZahra Faghani[1], Ahad Alizadeh[2] and Mohammadreza Barzegartahamtan[3]

[1] Department of Biostatistics, Faculty of Health, Mazandaran University of Medical Sciences, Sari, Iran
[2] Department of Epidemiology and Reproductive Health, Reproductive Epidemiology Research Center, Royan Institute for Reproductive Biomedicine, ACECR, Tehran, Iran
[3] Department of Radiation Oncology, Firoozgar Hospital, Iran University of Medical Sciences, Tehran, Iran

Corresponding author
Mohammadreza Barzegartahamtan,
Barzegartahamtan.mr@iums.ac.ir,
mbtahamtan@gmail.com

## ABSTRACT

**Purpose.** One of the characteristics of Prostate-Specific Antigen (PSA) is PSA slope. It is the rate of diminishing PSA marker over time after radiotherapy (RT) in prostate cancer (PC) patients. The purpose of this study was to evaluate the relationship between increasing RT doses and PSA slope as a potential surrogate for PC recurrence.

**Patients and Methods.** This retrospective study was conducted on PC patients who were treated by radiotherapy in the Cancer Institute of Iran during 2007–2012. By reviewing the records of these patients, the baseline PSA measurement before treatment (iPSA), Gleason score (GS), clinical T stage (T. stage), and periodic PSA measurements after RT and the total radiation dose received were extracted for each patient separately. We used a Bayesian dose-response model, analysis of variance, Kruskal–Wallis test, Kaplan–Meier product-limit method for analysis. Probability values less 0.05 were considered statistically significant.

**Results.** Based on the D'Amico risk assessment system, 13.34% of patients were classified as "Low Risk", 51.79% were "Intermediate Risk", and 34.87% were "High Risk". In terms of radiation doses, 12.31% of the patients received fewer than 50 Gy, 15.38% received 50 to 69 Gy, 61.03% received 70 Gy, and 11.28% received more than 70 Gy. The PSA values decreased after RT for all dose levels. The slope of PSA changes was negative for 176 of 195 patients. By increasing the dosage of radiation, the PSA decreased but these changes were not statistically significant ($p = 0.701$) and PSA slope as a surrogate end point cannot met the Prentice's criteria for PC recurrence.

**Conclusion.** Significant changes in the dose-response relationship were not observed when the PSA slope was considered as the response criterion. Therefore, although the absolute value of the PSA decreased with increasing doses of RT, the relationship between PSA slope changes and increasing doses was not clear and cannot be used as a reliable response surrogate endpoint.

## INTRODUCTION

Prostate cancer is an important health problem in men (*Esfahani, Ataei & Panjehpour, 2014*; *Wilt & Ahmed, 2013*). Also, it is reported as the most common malignancy and the second leading cause of cancer related death in men in many parts of the world (*Bidgoli, Jabari & Zavarhei, 2014*; *Obort, Ajadi & Akinloye, 2013*). Radiotherapy plays an essential role in the treatment of prostate cancer patients. Both the PSA amount and its changes over time, including velocity, density, and doubling time, are important in assessing response to RT (*D'Amico et al., 2003*; *Molenberghs et al., 2002*). Some of PSA metrics (e.g., PSA velocity greater than 1.5 ng/ml/yr and PSA doubling time <6 months) are surrogate for PC mortality or overall survival (OS) (*D'Amico et al., 2003*) but others (e.g., PSA decline ≥ 30% and PSA doubling time >12 months) are not surrogate endpoint (*Collette, Buyse & Burzykowski, 2007*; *Halabi et al., 2013*; *Valicenti et al., 2006*). Another of these changes over time is the PSA slope. In fact, it is the rate of PSA change over time after RT and has been previously discussed in the literature (*Vollmer & Montana, 1999*; *Suzman et al., 2015*; *Bellera et al., 2008*). Biochemical failure (BF) has various definitions such as two or three PSA rises, post-nadir increase to ≥ 3 ng/ml above the nadir and PSA value to be greater than a fixed cutoff level. Rising PSA was considered in these definitions (*Takamiya et al., 2003*) but PSA slope was used as a continuous variable or categorized outcome for determining disease-free survival in a few studies. *Proust-Lima et al. (2008)* assessed the relationship between prognostic factors, PSA dynamics and clinical failure (CF) using a complex two-stage model. They used a linear mixed model for prediction of PSA evolution in three phases after RT. *Takamiya et al. (2003)* found that a zero PSA slope in post treatment PSA supports cure of patients with long-term follow up after RT, but there are few studies that demonstrated the surrogacy of PSA slope for CF end point.

Some studies have suggested that biochemical responses such as freedom from BF would improve with increasing doses, but the freedom from CF or OS of patients did not change significantly (*Budäus et al., 2012*; *Wolff et al., 2009*; *Al-Mamgani et al., 2008*; *Peeters et al., 2006*). Also, dose escalation advantages and dose effect for low, intermediate and high risk groups were reported different in literature (*Pollack et al., 2002*; *Kuban et al., 2008*; *Zietman et al., 2005*) and there is controversy about the dose escalation benefits in the various risk groups especially low risk patients. Further research in this field will probably clarify these differences. Also, it is important to analyze the PC clinical relapse (as a true endpoint) and a within-subject response PSA slope (as a potential surrogate endpoint) of dose-escalated therapy (>70 Gy) in a population-based cohort.

Some studies have shown that radiation dose escalation is related to the risk of recurrence, treatment response and OS by using a joint modeling approach (*Taylor & Wang, 2002*; *Prentice, 1989*). However, the impact of radiation dose escalation on PSA slope is not clear. Another study (*Alizadeh, Mohammadpour & Barzegar, 2013*) showed that the PSA slope was related to the recurrence onset time. Different statistical methods such as mixture model, exponential model and Baysian model are used to investigate the relationship between dose and response in medicine. The dose–response Bayesian model tries to identify a connection between the average responses at nearby doses (*Ntzoufras, 2009*). We

consider the PSA slope as a treatment response and estimate the posterior mean. Working within a Bayesian framework avoids many of the implicit assumptions such as small sample size that restrict the validity of classical likelihood methods. For instance, most data sets used for dose–response analysis are very small, containing only a few dose groups with a few exposed subjects. In these situations, in spite of complexity in Bayesian model, Markov Chain Mont Carlo (MCMC) sampling method is quite effective at handling complex models and provide a clear advantage over Maximum-Likelihood estimation (MLE) (*Hamra, MacLehose & Richardson, 2013*; *Leininger, 2009*). The present study investigates the impact of radiation dose escalation on the PSA slope after radiotherapy by using a statistical Bayesian model. If the result of this study indicates that dose is a worthy predictor of PSA slope, then Prentice's criteria (*Prentice, 1989*; *Heller, 2015*) for PSA slope surrogate for the recurrence time of PC are evaluated.

## MATERIALS AND METHODS

### Patients

This historical cohort study was performed after getting approval from the ethical committee of Mazandaran University of Medical Sciences (IR.MAZUMS.REC.1394.1347). For this study, all localized prostate cancer patients who were treated by RT in the Cancer Institute of Iran (Tehran) from 2007 to 2012, were investigated retrospectively. The study data was taken from existing files of previously treated patients and the subject of consent form was not applicable (Data S1, Data S2).

By reviewing the records of these patients, the baseline PSA measurement before treatment (iPSA), Gleason score (GS), clinical T stage (T. stage), and periodic PSA measurements after RT and the total dose of radiation received were extracted for each patient separately. All the patients were classified into risk assessment groups. The stratification of patients was done with the use of D'Amico system as follows: patients with all conditions iPSA $\leq 10$ AND GS $\leq 6$ ANDT.stage $= T1 - T2a$ into Low-Risk, with a condition of PSA $= 10 - 20$ AND/OR GS $= 7$ AND/ORT.stage $= T2b$ into Intermediate-Risk, and with a condition PSA $> 20$OR GS $= 8 - 10$ ORT.stage $\geq T2c - T3$ into High–Risk prostate cancer (*Rodrigues et al., 2012*; *D'Amico et al., 1998*). Also, all patients were divided into four groups in terms of the total dose received as the main variable (<50 Gy, [50 70) Gy, =70 Gy and 70<Gy). After obtaining the pre-treatment and post-treatment values, the slope of PSA changes was calculated for each patient. In this study, PSA slope is the slope of the linear regression of PSA repeated measurements vs. time and was calculated by slope function in Excel software. Finally, the relationship between the radiation dose received and the slope of PSA changes was investigated.

**Statistical method**: In this study, we used a Bayesian dose–response model for analysis. In this model, the slope of PSA changes was calculated and considered as treatment response of the j$^{th}$ individual at the i$^{th}$ dosage level as follows:

$$slope_{ij} \sim Normal\left(\mu_i, \sigma_{slope}^2\right)$$

$\mu_i =$ the mean response for the i$^{th}$ dosage level, and $\sigma_{slope}^2 =$ the common variation at all dose levels.

Prior distributions:

$d_i$ = square root of the distance between the current and previous dosage levels,

$\sigma_\mu^2$ = the priori variance of the mean response at each dosage level (*Leininger, 2009*).

$$\mu_i \quad \sim \text{Normal}\left(\mu_{i-1}, d_i \sigma_\mu^2\right)$$
$$\sigma_{\text{slope}}^2 \sim \text{Gamma}$$
$$\sigma_\mu^2 \quad \sim \text{Gamma}$$
$$\mu_0 \quad \sim \text{Normal}\,(0, 0.0001)$$

A statistical software package that uses Markov Chain Monte Carlo, the BOA package of R software (available from http://CRAN.R-project.org/; *Smith, 2007*) trace plot, boxplot and ANOVA-type diagnostic test were used for analysis and interpretation of the above mentioned data. Written R code was explained in the supplemental file (RcodeBOAlinkeS3 and DataDescriptionS4).

The method of analysis of variance was used to compare the mean values of continuous variables, and the non-parametric Kruskal–Wallis test was used to compare the median values of PSA slope among the dose levels. The Kaplan–Meier product-limit method was used to estimate probabilities of time to recurrence or survival time, both measured from the start of RT. The log–rank test was used to compare distributions. Probability values less 0.05 were considered statistically significant.

## RESULTS

In this study, 195 localized prostate cancer patients who were treated by RT in the Cancer Institute of Iran between 2007 and 2012 were identified. The shortest and longest durations of follow-up were four and 67 months, respectively. The mean follow-up duration was 19.60 months. In terms of risk grouping, 13.34% of the patients were classified as "Low Risk", 51.79% were "Intermediate Risk", and 34.87% were "High Risk". The slope of PSA changes was negative for 176 patients, meaning that for most of the patients, the overall trend of PSA trajectory was descending. Table 1 shows the frequency distribution of patients in each radiation dose received level both in non-stratified and stratified patients. The PSA slope after RT by the follow-up period is illustrated in Fig. 1. The results of this study indicated that the long interval from the baseline PSA before RT to end of follow-up time was not correlated with a lower slope ($p = 0.98$). The maximum number of PSA measurements for patients after RT was 19 and correlation between follow-up duration and number of PSA measurements was statistically significant ($r = 0.77$, $p < 0.01$).

Table 2 lists the posterior parameters, including posterior mean, posterior median, posterior standard deviation, and posterior credible interval (2.5% and 97.5% percentiles), based on 12,000 iterations in Post RT patients in a regression normal model.

Figures 2A–2D is a boxplot that compares the posterior mean parameter of PSA slope in the dose levels and risk groups. The boxplots are arranged in ascending order. This figure demonstrated that PSA decreased after RT for all doses. When the radiation doses were increased, the post-RT PSA slope dropped further. The greatest reduction in the PSA slope

**Table 1** Frequency distribution (%) of prostate cancer patients in the non-stratified and stratified based on risk and radiation dose level.

| Groups | < 50 Gy | [5070) Gy | = 70 Gy | 70 < Gy | Total |
|---|---|---|---|---|---|
| **Without stratification** | 24 (12.31) | 30 (15.38) | 119 (61.03) | 22 (11.28) | 195 |
| **Low risk** | – | 3 (11.54) | 20 (76.92) | 3 (11.54) | 26 (13.34) |
| **Intermediate risk** | 8 (7.92) | 15 (14.85) | 64 (63.37) | 14 (13.86) | 101 (51.79) |
| **High risk** | 16 (23.53) | 12 (17.65) | 35 (51.47) | 5 (7.35) | 68 (34.87) |

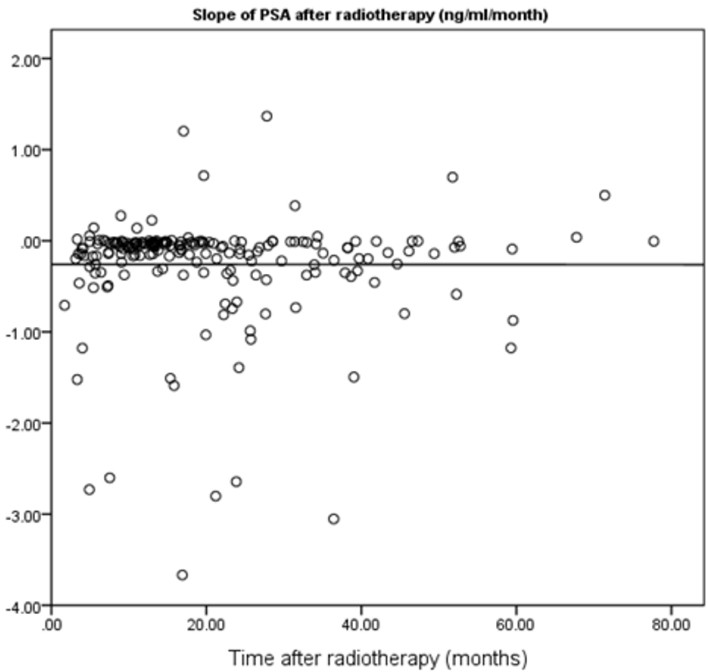

**Figure 1** Scatter plot for correlation between follow-up period and PSA slope after RT.

**Table 2** Posterior estimations of PSA slope means in a Bayesian model; without stratification.

| Dose level | mean | Standard deviation | 2.5% percentiles | Median | 97.5% percentile | Gelman–Rubin statistic |
|---|---|---|---|---|---|---|
| < 50 Gy | −0.1543 | 0.0654 | −0.2909 | −0.1509 | −0.0323 | 1.0005 |
| [5070) Gy | −0.2733 | 0.0775 | −0.4278 | −0.2723 | −0.1275 | 1.0013 |
| 70 Gy | −0.2203 | 0.0379 | −0.2952 | −0.2201 | −0.1464 | 1.0001 |
| 70 < Gy | −0.332 | 0.0903 | −0.5086 | −0.3319 | −0.1537 | 1.0003 |

was recorded for doses greater than 70 Gy. By increasing the dosage of radiation, PSA is decreasing at a faster rate, but these changes were not statistically significant and the dose response curve became a straight line for the total PSA slope mean in the all doses levels.

## The surrogacy testing by the Prentice's criteria

The Prentice criteria are the following. (I) Treatment is predictor for true endpoint. (II) Treatment is predictor for surrogate endpoint. (III) Surrogate is correlated with true

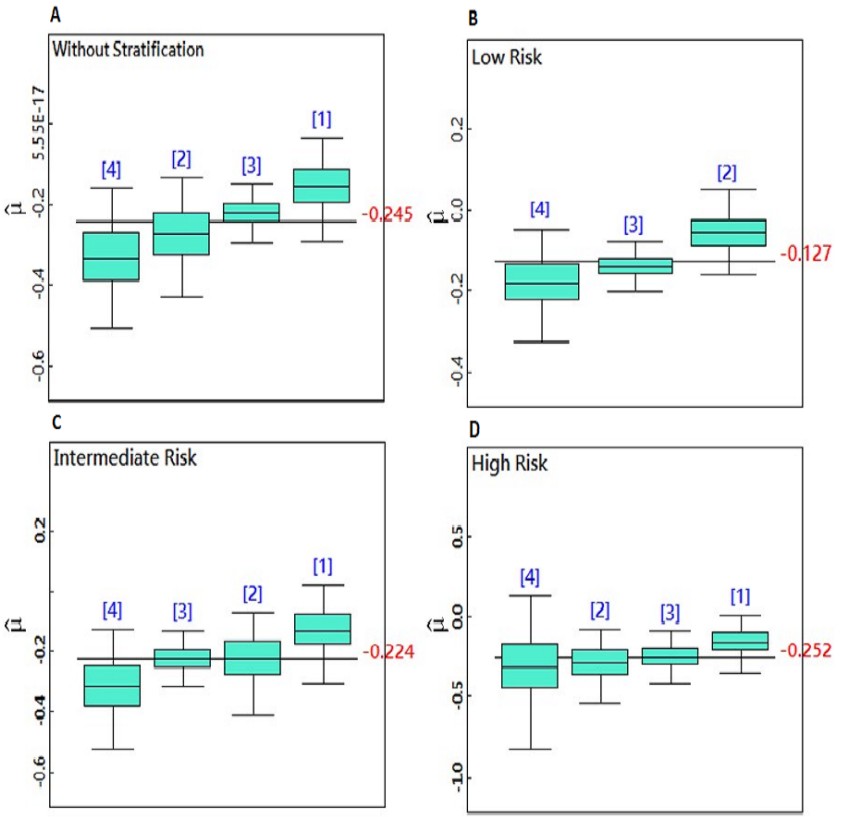

**Figure 2** (A–D) Boxplots for comparison of the dose level posterior means in the non-stratified and stratified patients.

endpoint. (IV) The full effect of the treatment on true endpoint is explained by surrogate endpoint.

Dose level was a statistically significant predictor of disease-free survival (Fig. 3A) and was satisfied in first condition of surrogacy. Mann–Whitney Test for total dose by recurrence event shows that the mean-ranks of doses is different in two groups with or without recurrence ($p < 0.001$).

As Table 2 indicates, the Bayesian credible interval among four dose groups is overlapped and median of PSA slope in groups is same. There was no significant difference among four groups in PSA slope mean ($p = 0.705$). Also, the distributions of slope is the same across categories of total dose by Kruskal–Wallis Test ($p = 0.902$). Consequently, total dose was not a worthy prognosticator for PSA slope as a continuous outcome; because of this, the Prentice's second criterion was not met.

PSA slope was a statistically significant predictor of time to PC recurrence (disease-free survival). The means of disease-free survival time in patients who had positive and no positive PSA slope values were estimated 33.35 (95% CI [19.2–47.4]) vs. 59.6 (95% CI [52.8–66.4]) months respectively by using the Kaplan–Meier method (log rank test $p < 0.001$). Figure 3B demonstrates the effect of changes in PSA slope on disease-free

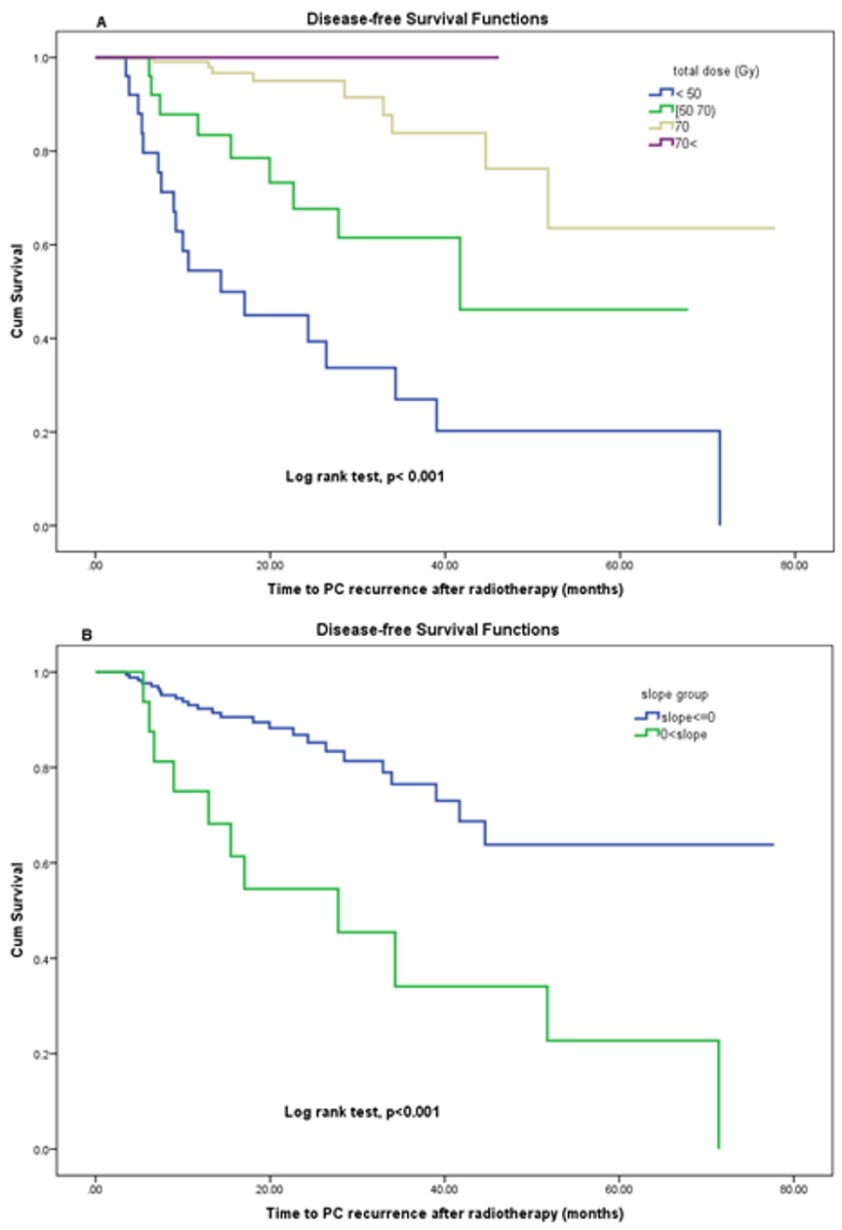

**Figure 3** (A, B) Disease-free survival function by (A) total dose levels and (B) PSA slope after RT.

survival using Kaplan–Meier analysis. The change in PSA slope (as a dichotomized variable) was significantly prognostic for disease-free survival (log rank test, $p < 0.001$) therefore the third criterion of Prentice was met. But the Mann–Whitney Test for slope by recurrence events shows that the mean-ranks of slope is not different in two groups with or without PC recurrence ($p = 0.345$). Consequently, PSA slope as a continuous outcome did not meet the third criterion of Prentice.

In a multivariate logistic model, with PSA slope and dose level as inputs to the model, the dose is remained statistically significant ($p = 0.007$), but the PSA slope was not significantly

related to recurrence ($p = 0.594$). Also (Figs. 4A, 4B) demonstrates the effect of total dose on disease-free survival when PSA slope values are zero or less (no positive) vs. positive PSA slope group by using Kaplan–Meier analysis. The change in radiation total dose was significantly prognostic for disease-free survival in the first group (log rank test, $p < 0.001$) but not in the second group (log rank test, $p = 0.9$). This result indicates that patients with negative or zero PSA slope had significantly PC recurrence less than others if they had been previously treated with higher doses rather than low doses. Thus, PSA slope as a surrogate endpoint cannot fully explain the effect of dose and the Prentice's criterion 4 was not met for capturing total effects of radiation dose by the PSA slope on PC recurrence.

## DISCUSSION

The prostate-specific antigen is a biomarker and its changes can be used as a surrogate endpoint for response evaluation in prostate cancer patients (*D'Amico et al., 2003*). In our study, the slope of PSA biomarker was intended as a surrogate endpoint for PC recurrence. The PSA slope is defined as the rate of PSA change over time following RT. For PSA slope calculation, there is a controversy regarding the starting time and end of follow-up time. *Anwar et al. (2014)* calculated PSA slope as outcome after radiotherapy for 3 intervals following RT (0 to 1 year, 0 to 2 years, and 0 to 3 years) and reported that the median PSA slope for conventionally-fractionated external beam radiotherapy (CF-EBRT) was $-0.09, -0.04, -0.02$ ng/ml/month, for durations of 1, 2 and 3 years post RT. Similarly, for stereotactic body radiotherapy (SBRT), the median PSA slopes were $-0.09, -0.06, -0.05$ ng/ml/month. The PSA slope for SBRT was greater than CF-EBRT ($p < 0.05$) at 2 and 3 years following RT, although similar during the first year. PSA evolution following RT was described by Prosto Lima et al. with three components: post-therapy level, short-term decline with exponential function of time and long-term PSA rise as a linear function of time. *Takamiya et al. (2003)* calculated PSA slope for each patient with starting time 3 years after RT. *Antonarakis et al. (2012)* compared the pre-study entry (pre-treatment) PSA slope and post-treatment PSA slope to describe the association between PSA kinetic and metastasis-free survival. Although the connection between pre- and post- treatment PSA kinetic may clarify the role of PSA slope as a predictor or response variable in various studies. But in the retrospective studies, mostly the frequency of pre-treatment PSA evaluation is not regulated. Therefore, pre-treatment PSA slope in majority of patients cannot be computed. For ease of applicability, we included the last pre-treatment PSA value into serially PSA measurements after RT for each patient. By incorporating the pre-treatment PSA value directly into the slope calculation, a time-dependent covariate can be joined to post-treatment PSA behavior. Also, it takes into account the PSA change during treatment and reduces the time dependent interval censoring bias.

While some studies have shown that there is a relationship between the magnitude of the PSA decline and clinical outcomes such as: development of distant metastases, risk for relapse (*Kaplan, Cox & Bagshaw, 1991*; *Chauvet et al., 1994*) and metastasis-free survival (*Antonarakis et al., 2012*), other studies have not shown this relation (*Ritter et al., 1992*; *Zagars & Pollack, 1993*; *Zagars & Pollack, 1997*). In our study, with increasing dose level, the posterior mean of the PSA slope dropped with a faster rate in most patients (176 out

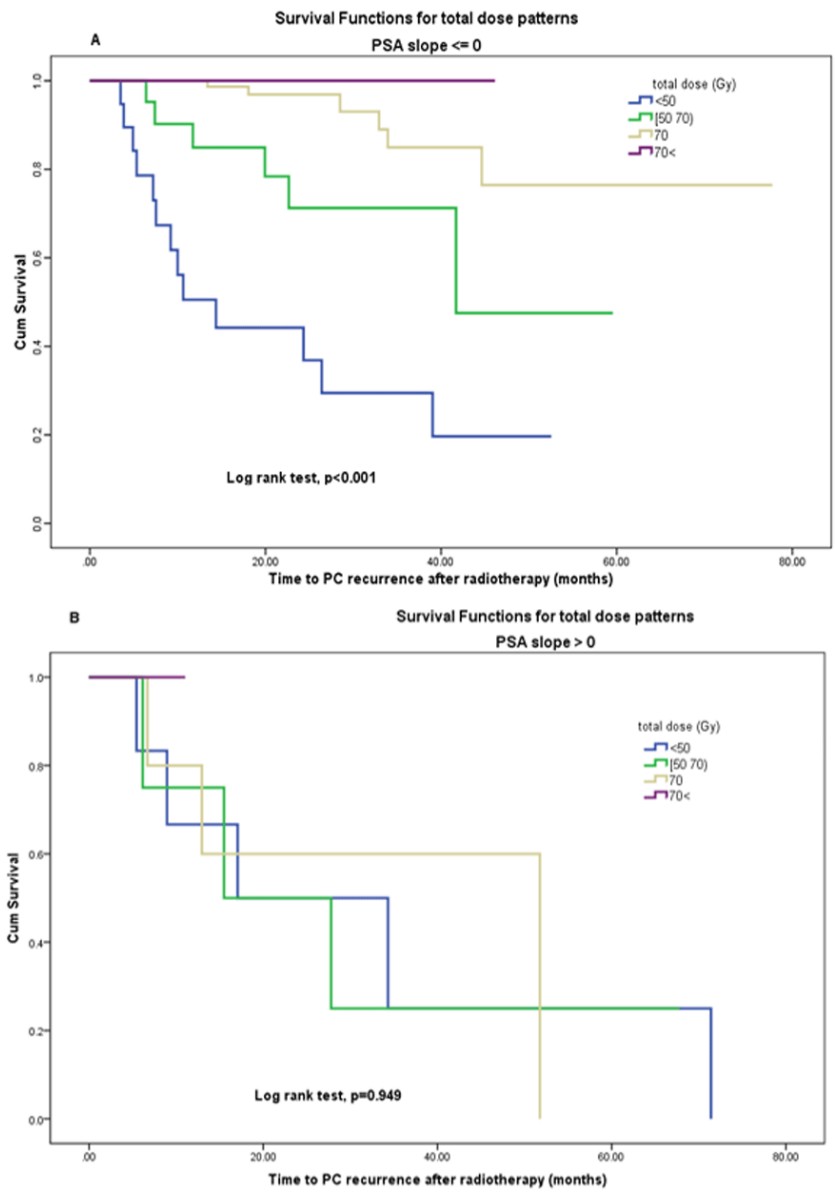

**Figure 4** (A, B) Disease-free survival function by (A) total dose levels in PSA slope ⟵ 0, (B) total dose levels in PSA slope > 0.

of 195, = 90%) and disease-free survival time in these patients was increased. This means that although there is better therapeutic response with higher doses of RT in prostate cancer, the PSA slope changes as a mediate variable cannot fully save the dose effect on true response. Of course, this does not necessarily mean more benefit to the patient, because other endpoints such as early and late complications, metastasis-free survival and OS must be studied as well.

Some studies have suggested that biochemical responses would improve with increasing doses, but the CF or OS of patients did not change significantly and the acute and late

gastrointestinal and genitourinary toxicity complications also increased (*Wang et al., 2014*; *Zietman et al., 2005*; *Al-Mamgani et al., 2008*; *Peeters et al., 2006*). The initial studies were performed with a three dimensional (3D) technique. Further studies showed that advanced methods of radiotherapies for PC, such as Intensity modulated radiation treatment (IMRT), were associated with fewer side effects, although they did not increase OS (*Budäus et al., 2012*; *Wolff et al., 2009*). In another randomized dose escalation, 301 patients were investigated in terms of doses of 78 Gy and 70 Gy. Pollack et al. showed that an increase of 8 Gy resulted in a highly significant improvement in freedom from BF (70% and 64%, respectively ($p = 0.03$)) for patients at intermediate to high risk, which is consistent with our study in freedom from recurrence (*Pollack et al., 2002*; *Kuban et al., 2008*). Overall survival improvement and advantage for dose escalation from 64.8 Gy to 86.4 Gy were reported in all risk groups (*Zelefsky et al., 2011*; *Hall et al., 2015*) but *Pollack et al.*'s (*2004*) study has not demonstrated dose-escalation advantages for low-risk cases. Another study (*Shelan et al., 2013*) strongly supported the application of at least 70 Gy rather than 66 Gy. The results of our study revealed that however doses greater than 70 Gy decreased PSA more than other dose levels (<50, [50 70), 70 Gy) but significantly increased disease-free survival time similar to above studies.

Another point about this study is that with using a simple approach, as linear regression, the last measurement before treatment and the pattern of PSA after treatment, were linked. Linear regression played the role of a flattening function. In fact, instead of a nonlinear function, the slope of the PSA as a smoothing function causes flexibility in the model. When PSA variability within subjects is important and the true end points are related to complex biological process, the values of PSA should not be relied on in the short term. Then long term follow up is needed for the PSA change point detection and two-component mixture model, exponential model and other approaches for tumor kinetic are suggested (*Proust-Lima et al., 2008*).

In this paper, our focus was on the use of a potential surrogate biomarker for a Bayesian analysis of dose–response relationship. It was done by a retrospective review of the outcomes in prostate cancer patients. This feature limited the results of our study, so that our findings were restricted to patients with minimum time to clinical recurrence or lost to follow-up without recurrence as censored data. Short follow-up time will not allow us to determine the future disease progression or others true end points after recurrence. Since there is increasing interest in the use of surrogate marker endpoints in trials, further prospective clinical studies will be required in the field.

## CONCLUSION

Significant changes in the dose–response relationship were not observed when the PSA slope was considered as the response criterion. Despite the fact that PSA slope was prognostic factor for disease-free survival in this study, the association between PSA slope changes and dose increase was not clear and therefore it cannot be used as a reliable surrogate for a PC recurrence endpoint.

## ACKNOWLEDGEMENTS

We thank all those who have helped us in collecting patient data, This article is part of a master's thesis for Ms. Z. Faghani and registered (no. 1347) in the Deputy of Research in the Mazandaran University of Medical Sciences.

### Funding
The authors received no funding for this work.

### Competing Interests
The authors declare there are no competing interests.

### Author Contributions
- Reza Ali Mohammadpour, Ahad Alizadeh and Mohammadreza Barzegartahamtan conceived and designed the experiments, performed the experiments, analyzed the data, contributed reagents/materials/analysis tools, prepared figures and/or tables, authored or reviewed drafts of the paper, approved the final draft.
- Jamshid Yazdani- Charati conceived and designed the experiments, performed the experiments, prepared figures and/or tables, authored or reviewed drafts of the paper, approved the final draft.
- SZahra Faghani performed the experiments, analyzed the data, contributed reagents/materials/analysis tools, prepared figures and/or tables, authored or reviewed drafts of the paper, approved the final draft.

### Ethics
The following information was supplied relating to ethical approvals (i.e., approving body and any reference numbers):

This study was performed as a historical cohort after approving in ethical committee of Mazandaran University of medical sciences (IR.MAZUMS.REC.1394.1347).

### Data Availability
Raw data is available as Supplemental Files.

### Supplemental Information
Supplemental information for this article can be found online at http://dx.doi.org/10.7717/peerj.7172#supplemental-information.

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
