# Peer review of "Radiation dose-response (a Bayesian model) in the radiotherapy of the localized prostatic adenocarcinoma: the reliability of PSA slope changes as a response surrogate endpoint"

_PeerJ, doi:10.7717/peerj.7172_

## Round 0.1 · original submission · Major Revisions

The paper has been reviewed by three experts and all three identified a number of concerns that will need to be addressed before the study is suitable for publication in PeerJ.

·

Basic reporting

The authors seek to correlate radiotherapy dose levels with PSA kinetics after therapy through a Bayesian approach. The authors call such PSA kinetics a "surrogate endpoint" but it is unclear for what endpoint they are seeking a surrogate. They rightly point out in the discussion that use of PSA velocity as a surrogate is controversial: without measurements of some endpoint in this study, the authors are unable to apply Prentice criteria and claim PSA slope as a surrogate in this treatment setting. If any follow-up data is available for these patients, analytic inclusion would greatly improve the paper.

The study would benefit from explaining how the current study complements the following studies, which address highly similar hypotheses (PSA kinetics after radiation therapy):
https://www.ncbi.nlm.nih.gov/pmc/articles/PMC2757130/
https://www.ncbi.nlm.nih.gov/pubmed/19825885

The raw data is in Excel format, and the columns appear in strange locations of the .xlsx file. It might be easier to instead share a .csv/.txt/similar.

Experimental design

The methods for calculating PSA slope and how the authors' calculation differs from existing notions of "PSA slope" and "PSA velocity" is unclear. To quote the paper:

"Another of these changes over time is the PSA slope. It differs from the above concepts and is another concept. In fact, it is the rate of PSA marker diminishing over time after radiotherapy."

The authors helpfully cite Suzman et al. 2015, where we get the following definitions:
* PSA velocity is the slope of the linear regression of PSA vs time
* PSA slope is the the slope of the linear regression of log(PSA) vs time
You can (and several authors have) compute either of these quantities before or after treatment, so claiming that a new concept is created here is inaccurate (if only timing with respect to treatment delivery defines the novelty). Related to the above definitions, it's not clear how the slopes were calculated in this study. Was PSA log-transformed? If not, why not?

From the paper:
"To avoid complexity in dose response models, the Bayesian model provides a flexible approach for accurate detection of the dose response curve."
- Can the authors describe how using a Bayesian model avoids complexity versus, for example, a simple linear regression as other authors have used?

Broadly, the details of the Bayesian dose-response model are unclear. Why is one needed? I think I see a prior distribution specified, but it's not labeled as such.

How were the cutpoints to create the four radiation dose categories chosen?

A helpful guidepost for the authors to consider when revising: could someone take the raw data and exactly reproduce the numbers (p-values etc) shown in the paper by following steps written in the methods section? As written now, I don't believe this is feasible.

Validity of the findings

The sample size is rather small, particularly within dose groups, so concluding that "the PSA slope changes association with dose increase was not clear and therefore it cannot be used as a reliable response surrogate endpoint" is not justified: lacking statistical significance in this study may simply be the result of low power. Changes in PSA kinetics as a result of varying radiotherapy doses have been robustly shown in larger cohorts (for example those I cited earlier in the review). The authors could calculate how much of a change in PSA slope their study was powered to detect.

There is very sparse data for the low-risk patient group within dose levels: showing boxplots in Figure 2 for the distribution of 3 points is inappropriate (this happens for the two dosage groups). Simply plotting the points might help resolve.

Reviewer 2 ·

Basic reporting

The writing is mostly clear and unambiguous. However, there are certainly sentences that I suggest the authors reword or clarify (see Grammatical Comments). I have also listed some minor grammatical errors. The introduction contains relevant background information and references. The structure of the paper seems appropriate and figures are relevant to the content of the article. The raw data can be opened. However, I would suggest including a key to explain this data (see comment #7 below). This study seems self-contained.

Experimental design

The work seems original. However, as mentioned below, I suggest the authors explain how this work differs to references given in the paper (see comment #5). The research question is clearly defined (What is the relationship between PSA slope and increasing radiotherapy doses?) and meaningful. Various changes in PSA characteristics are important in assessing response to radiotherapy. However, according to this paper, the impact of radiation dose escalation on PSA slope is not clear and thus is of interest to study. The methods of the investigation seem appropriate and ethical. However, I would suggest the authors add more detail in order to be reproducible by another investigator.

Validity of the findings

The data in this study were from a previous study. The statistical analysis of the data seems appropriate (see comment #2), although I recommend that more details be provided. The conclusion is stated clearly and is connected to the original question.

Additional comments

Below I list some specific comments.

Comments about content:
1. Line 118. The authors divide the patients into four groups in terms of the total dose received. What is the standard dosage for radiotherapy? How were these categories chosen? Why did the authors choose a category exactly for 70 Gy? Why is 71 Gy not in a category? Do no patients have a dosage level of 71 Gy?
2. How exactly do you calculate the slope? It seems that, for example, two patients might have the same PSA values (initial PSA and PSA after RT) but one patient was able to get their PSA measurement after RT earlier than the other. That would indicate that the first patient has a more negative (steeper) slope, when in fact, they may have responded the same. Does this cause a problem in the study?
3. Figure 1.
- Why aren’t there 195 circles? Doesn’t each circle represent a patient? If not, the authors should clarify what this figure represents.
- Line 145 states that this figure is the mean of the PSA slope between the baseline pretreatment PSA and the first PSA measurement after RT. What are the authors averaging over?
- If I am understanding this figure correctly, the authors could add more information into Figure 1 by having different shapes or colors to represent the different groups (risk, radiation dosage, etc.).
4. Lines 146-147. The authors state that this indicated that the long interval to the first treatment after radiotherapy was accompanied by a slower slope. Isn’t this what you might expect? (See comment #2).
5. Lines 165-168. How exactly does this study differ to the ones mentioned?
6. Lines 178-181. Can the authors cite these studies?
7. I suggest the authors create a key for the raw data. For example, what is “HT”? Does “year” mean age of the patient?, Does “No-mon of fallow” mean number of months of follow-up? Why are there missing values in this column? etc.

Grammatical Comments
8. Line 49. The authors write RT before defining it. (It is defined on line 84).
9. Line 89. I think the period between “radiotherapy. (Vollmer…” should be removed.
10. Line 116, I think there should be a space between “orT.stage”.
11. Line 147. I don’t think it is mathematically correct to say a slope is slower. Perhaps “the PSA is decreasing at a slower rate”, or “a smaller slope in magnitude”. Similarly with line 203 (“slope is faster…slower”).
12. Lines 147-149. This sentence is a little confusing. Perhaps the authors can reword it.
13. Lines 173-177. These lines are not exactly clear. What exactly do you mean by “The PSA slope dropped with a slow rate”? How does this sentence mean that although there is a better therapeutic response with higher doses of radiotherapy, the PSA slope changes are not significant? And why would the fact that PSA slope changes are not significant mean more benefit to the patient?
14. Lines 186-187. I suggest the authors define the abbreviations.
15. Line 188. The “an” should be removed in “In an another randomized…”
16. Line 210-213. Do the authors mean “…our findings were restricted to patients without recurrence and with a short follow-up time”, or “…our findings were restricted to patients without recurrence and without a short follow-up time”? If it is the former: In lines 203-206, the authors state that outcome should not be relied on the values of PSA in the short term. Doesn’t this mean that your study isn’t reliable? If it is the latter, then why is a long term follow up needed (line 205)?
17. There are different fonts throughout the paper.
18. Table 2. In the second row, the dose level reads ≥50 Gy <. I think there should be a 70 after the <.
19. Figure 2. Some words and symbols are in bold and others are not. I would suggest being consistent.

Reviewer 3 ·

Basic reporting

ABSTRACT
1. The sentence beginning with “It is different from the PSA amount…” should be rewritten for clarity.”
2. It is stated that the PSA changes were negative for 176, but the total number of patients is not previously mentioned.

INTRODUCTION
1. It would be beneficial to add more details to the Introduction. The authors touch on one specific model that studied the relationship between PSA slope and time to recurrence, however the examples presented in the Discussion (lines 184-202) might help to motivate the study better in the Introduction.
2. Additionally, the authors mention that “different statistical methods are used to investigate the relationship between dose and response in medicine,” however they do not state specific examples besides the Bayesian model. What are the other methods that could have been used to study this concept and why was this model used over other methods? Referencing additional methods or background on the Bayesian method would be useful due to the few details included in the statistical method subsection.

RESULTS
1. Was there a correlation between positive PSA slope and dosage or risk-stratification?
2. The second sentence of the second paragraph of the results section (lines 145-149) should be rewritten for clarity. The data in Figure 1 shows an inverse relationship between PSA reduction and follow-up time, but this is not clear from the text.
3. The dose-response curve is mentioned (line 158), but is not shown in the manuscript.
4. For Figure 1, it may be useful to differentiate doses to show relationship or lack or relationship between dose and PSA slope. It may also help explain why those 19 patients saw an increase in their PSA, rather than a decrease.
5. For Figure 2, results should be shown on same scale for comparison purposes.
6. What do the red numbers, as well as the horizontal line, signify in Figure 2?
7. It is mentioned throughout the results that the differences were not significant (lines 158, 175) however it would be beneficial to include the p-values that lead to these conclusions both in Figure 2 and in the text.

DISCUSSION
1. Although the paper referenced on line 166 fit a model of tumor kinetics, no relationships between PSA slope and clinical outcomes were described.
2. It is mentioned on lines 178 through 180 that previous studies suggested that response would improve with increasing doses but references are not provided until three lines later.
3. The results of Pollack et al. are said to be consistent with the authors’ study, but results presented earlier in the text do not support this statement. Though increasing the dose does show an increase in the PSA reduction, it has been said not to be statistically significant (lines 158 and 175). If the p-values do indeed show a significant difference, these values should also be reported on line 202.
4. As previously mentioned, it may be beneficial to move the discussion of prior studies (lines 184-202) into the Introduction.


Minor remarks:
1. Lines 87-89 should be combined into one sentence.
2. On line 95, it states “The present study shows the impact of radiation dose…” but a clear relationship is not shown throughout the manuscript. Consider changing “shows” to “investigates.”
3. The stratification of the patients based on the D’Amico system is not clear and consistent. Line 113 should state “iPSA <= 10 OR GS <= 6 OR T.stage = T1-T2a,… 10 < iPSA <= 20 OR GS = 7 OR T.stage = T2b,… PSA > 20 OR GS = 8 OR 10 OR T.stage = T2c-T3" for consistency.
4. In Table 2, the “dose levels” label should be moved to top row with the others and line 4, column 2 should read [50,69) (or some other variation to show that 69 is the upper limit).
5. On Figure 2, “mu” should be changed to the Greek symbol for mu.
6. On line 174, omit “This means that”.
7. Double-check statistics shown on line 185 and include the p-value.

Experimental design

The authors aimed to study the relationship between PSA slope to study the dose-response relationship in a retrospective study. The methods were described sufficiently, such that results may be reproduced. However, the Introduction did not serve as a clear motivator for why this research was filling a necessary gap.

Validity of the findings

The findings of the study suggest that there is not a clear dose-response relationship when using PSA slope as a response criterion. However, conflicting results were presented throughout the text (line 192) and the results were not presented with appropriate statistical methods (p-value).

Additional comments

Overall, the findings of the study are interesting, but could benefit from further analysis to clearly prove or disprove that a relationship between dose and PSA slope exists. Additionally, the English language should be reviewed to ensure that the text is clear to all audiences.

---

## Round 0.2 · Major Revisions

From the below reviews you will see that the referees appreciate the edits that have been made, but still raised a number of concerns.

Reviewer 2 ·

Basic reporting

There are still numerous sentences throughout the paper that are hard to read due to grammatical errors. I have listed some of these sentences below.

The following phrases have grammatical errors or are difficult to read. I would recommend that the authors read through the paper again as I have not listed every grammatical error.
1. Sentence on line 90-93, “Both the PSA amount … surrogate end point”. This sentence is very long and hard to understand. I would recommend splitting it into two sentences.
2. Figure 1 title, “slpoe” should be “slope”
3. Figure 3, part of the legend is “…”
4. Figure 4, “gy” in the legend should be “Gy”
5. Lines 201 – 204, “As table 2 indicates, … Kruskal-Wallis Test”. These sentences are hard to understand and are grammatically incorrect.
6. Lines 214 – 216, “Although the changes in PSA slope…”. I am not sure what this sentence is saying.
7. Line 227, “were significantly less likely to involve with PC recurrence…”. What does “involve with PC recurrence” mean?
8. Line 252, it is still not clear to me what “PSA slope dropped with a slower rate in most patients” means. Doesn’t the slope become more negative with increasing dose level, in which case “PSA decreased at a faster rate”?
9. Line 273-275, “…revealed that however doses equal”. I am not sure what this is trying to say.
10. Line 277-278, the phrase “using a simplistic approach as linear regression to time indeed nonlinear functions” is confusing.

Experimental design

It is still not obvious to me that the results are reproducible by another investigator (see comments #1, 5 below).

Validity of the findings

See comment #5 below.

Additional comments

1. The improvements to the data headings make it easier to read. However, I would still recommend a key to explain the data. For example, there is a section called mean of psa. What is this averaged over? What is the difference between iPSA (baseline PSA measurement before treatment) and IniPSA? I assume this isn’t the raw data as there is a category for number of repeated PSA, but there isn’t the actual repeated PSA measurements. In order to reproduce the results, would we not need these values to calculate the slope?
2. Line 98, the authors state that BF has various definitions, but rising PSA was considered for all of them. Can the authors cite the “them”?
3. Line 109, why do dose escalation advantages and dose effect need to be investigated?
4. Line 123, I would recommend that the authors add something like “in this situation” unless the authors believe that it is always advantageous to use MCMC over MLE.
5. It is still not clear to me exactly how you calculate PSA slope. On line 147, the authors state that PSA slope is the slope of the linear regression of PSA repeated measurements vs. time and was calculated by the slope function in Excel. Are these repeated measurements in the excel file (see Comment #1)?
6. The authors should describe what j is on line 154.
7. Line 193, should “PSA is decreasing at a slower rate” be “PSA is decreasing at a faster rate”?
8. Line 235-256. In this paragraph, the authors state that there is controversy regarding the starting time and end of follow-up time for calculating the PSA slope. The authors then explain various studies. How do the author’s study fit into this?

Reviewer 3 ·

Basic reporting

The manuscript is mostly clear and unambiguous, however there are several areas that need to be rewritten for clarification (see general comments). The introduction and background show relevance, however they could benefit from some revision to improve the overall flow of the manuscript. On lines 92-93, the authors state that “some of these PSA metrics are surrogate…some of them are not surrogate end point.” Please provide specific examples of those that are. From line 115, what specific statistical methods are used to investigate the relationship between does and response?

For the raw data, it would be beneficial to include a legend for each column label.

The figures are relevant to the study. The authors state that “The purpose of this study was to evaluate the relationship between increasing RT doses and PSA slope as a potential surrogate for PC recurrence.” This point could be clearly driven by the data shown in Figure 1 if the authors differentiate between dose and PSA slope. In their response, they stated that they could not differentiate between doses. Why not?

The authors state in their response regarding Figure 2 that “This figure was corrected but we could not correct it.” Why not? This leads me to question the reproducibility of the study.

In Table 1, the dose group labels need to be revised for clarity. Perhaps using interval notation for the second group might be clearer (i.e [50,70)). The same should be done in Table 2. The table caption for Table 1 should also be revised.

Experimental design

The research question is well defined and relevant. The methods are described with sufficient detail and information to replicate.

The dose groups (lines 145-146) are not consistent with the abstract or Tables 1 or 2. The abstract has the doses ranging from 50 to 69 but the table seems to include any level below 70. Please revise. “Statistical method:” on line 151 should be bolded and lines 167-168 are not clear.

Validity of the findings

The authors have found that increasing dose results in a lower PSA slope, though the overall difference was not significant between dose groups. Lines 189-194 should be revised to clearly get this point across. Saying that PSA is decreasing at a “slower rate” contradicts their main findings. From line 179, “the overall trend changes were reduced” is not clear.

On lines 199-200, the authors state “the mean rank of dose is difference [different] in two groups with or without recurrence.” Should this be between the four groups? The final sentence of the results (lines 228-230) do not clearly flow from the previous results. How do the previous lines imply that the PSA slope cannot fully explain the effect of dose?

There is not a clear association of the first paragraph of the Discussion to the conclusions of the study. Did the authors find an issue in their calculation of PSA slope based on the interval used? The authors state that the follow-up time ranged from four to 67 months. Did this have an impact on the PSA slope? The second paragraph of the Discussion should be revised. In particular, lines 252-253. It is not clear to say that the PSA slope dropper with a slower rate. If the authors mean that the PSA slope is lower, then an increase in disease-free survival should be expected. The sentence following this, “This means that although…the PSA slope changes cannot fully explained [explain] it” does not clearly flow from the previous sentence and should be revised. On line 273, it’s stated “…doses equal to or greater than 70 Gy decreased PSA more than other dose levels but significantly increased free-disease survival time similar to above studies.” Using “but” would imply that this is a negative result. The “but” should be changed to “and”. Lines 277-281 are not clear.

Additional comments

Please see below for general grammatical comments:

Abstract
1. Line 42 should be revised to say “It is the PSA rate of change over time…”
2. Line 46 and line 134, change “during” to “from.”
3. Lines 49-51 are not clear.
4. The dose listed on line 54 is not consistant with the main body of the manuscript or Tables.

Introduction
1. Lines 94-97 should be revised for clarity.
2. Line 98 change “for all of them” to “in these studies.”
3. Line 102, change “investigated” to “found.”

Results
1. Please remove “First”, “Second”, “Third”, “Fourth” in the Results.
2. Line 202, “because the all of” and line 203, “significant differences there are among” should be revised.
3. Please use a consistent categorization of the slopes. For instance, on line 209, “positive and no positive” is used but then on line 223 “zero or less (decrease) vs. positive” is used.
4. Line 219 is not clear.
5. Line 220 should be revised to. Perhaps say “In a multivariate logistic model with PSA slope and dose level as inputs to the model…”

Discussion
1. Lines 234-235 should be revised since PSA slope can be zero, positive, or negative.

Figures
1. Typo in title of Figure 1 (slpoe).

---

## Round 0.3 · Minor Revisions

Thank you for your excellent improvements of your manuscript. In line with the reviewer , please correct the language of some of the important statements to understand the importance of the work.

·

Basic reporting

Further requests that follow from Reviewer 2's comment numbering:

## Basic reporting
1. This sentence remains unclear to me due to grammatical errors: "Only some of PSA metrics such as: PSA velocity greater than 1.5 ng/ml/yr and PSA doubling time <6 months are surrogate for PC mortality or overall survival (OS) (D’amico 2005) but others as PSA decline >=30% and PSA doubling time >12 months are not surrogate end point. (D’amico 2012, Collette 2008, Halabi et al 2013 and Valicenti et al 2006)."
5. Still not grammatically correct: "by using Bayesian model. The all of Bayesian credible intervals for all dose groups are overlapping."
6. This sentence remains redundant; additionally, a very brief description of what the 3rd (and other) Prentice criteria are could be helpful for this manuscript. The ordering of Prentice criteria are mostly consistent across the literature, but clarity would greatly assist the reader in this paper. "Because of this the third Prentice criterion was not met, consequently PSA slope as a continuous outcome did not meet the third criterion of Prentice."
7. Still unclear: does this sentence mean -- within the subset of patients with negative or 0 PSA slope, PC recurrence was significantly less frequent among patients with high doses compared to patients with low doses? "This result indicates that patients with negative or zero PSA slope had significantly PC recurrence less than others if they had been previously treated with higher doses rather than low doses."

A continuation of a request from Reviewer 3:
The following edited sentence in response to a lack of clarity in the discussion has grammatical errors that make it unclear what the authors are claiming. This is a critical point in the manuscript describing why the study is important, and the work would be improved if this section was easy to understand.
"Antonarakis et al (2012) has compared the pre- PSA slope and post- PSA slope. This issue may effects on the evaluation of PSA slope as a predictor or response variable in various studies. Incorporating pre-treatment PSA level in slope calculation indirectly joints a time-dependent covariate to post-treatment PSA behavior. Also it takes into account the PSA change at during treatment and reduce the time dependent interval censoring bias."

Experimental design

From Reviewer 2's comment to the author, item number 3: The response still didn't answer "why"

Validity of the findings

Broadly, some vagueness in the methods description remains. One solution would be to include code (the Statistical Method section implies that R was used) in the supplement alongside the data file. Ideally, this code would read and manipulate the .csv and/or .xlsx files therein and produce the reported numbers and figures directly. Such an effort would greatly improve the reproducibility and interpretability of this work.

Additional comments

The authors did a commendable job in responding to the previous round of reviews. The requested additions to the data file and figures are helpful. Some requests, primarily related to grammatical changes and clarity, were not fully addressed.

---

## Round 0.4 · accepted · Accept

We appreciate your effort in responding to the reviewers comments, which has significantly improved the manuscript.